# Antioxidant Properties of Hydrogen Gas Attenuates Oxidative Stress in Airway Epithelial Cells

**DOI:** 10.3390/molecules26216375

**Published:** 2021-10-21

**Authors:** In-Soo You, Subham Sharma, Ailyn Fadriquela, Johny Bajgai, Thuy Trinh Thi, Md. Habibur Rahman, Jaeyong Sung, Hwang-Un Kwon, So-Yeon Lee, Cheol-Su Kim, Kyu-Jae Lee

**Affiliations:** 1GOOTZ Co., Ltd., 79-6, Yuljeong-ro 247 beon-gil Yangju-si, Gyeonggi-do, Suwon 11457, Korea; igootz@naver.com (I.-S.Y.), kwon@mygootz.com (H.-U.K.), sylee@mygootz.com (S.-Y.L.); 2Department of Environmental Medical Biology, Wonju College of Medicine, Yonsei University, Wonju, Gangwon-do, Chuncheon 26426, Korea; subhamsharma047@gmail.com (S.S.); ailynfadriquela@gmail.com (A.F.); johnybajgai@yonsei.ac.kr (J.B.); tththuy@hpmu.edu.vn (T.T.T.); pharmacisthabib@yonsei.ac.kr (M.H.R.); cs-kim@yonsei.ac.kr (C.-S.K.); 3Department of Global Medical Science, Wonju College of Medicine, Yonsei University, Wonju, Gangwon-do, Chuncheon 26426, Korea; 4Department of Laboratory Medicine, Wonju College of Medicine, Yonsei University, Wonju, Gangwon-do, Chuncheon 26426, Korea; 5Department of Mechanical and Automotive Engineering, SeoulTech, 232, Gongreung-ro, Nowon-gu, Seoul 01811, Korea; jysung@seoultech.ac.kr

**Keywords:** hydrogen gas, oxidative stress, anti-oxidant, airway epithelium, MAPK signaling

## Abstract

Oxidative stress plays a crucial role in the development of airway diseases. Recently, hydrogen (H_2_) gas has been explored for its antioxidant properties. This study investigated the role of H_2_ gas in oxidative stress-induced alveolar and bronchial airway injury, where A549 and NCI-H292 cells were stimulated with hydrogen peroxide (H_2_O_2_) and lipopolysaccharide (LPS) in vitro. Results show that time-dependent administration of 2% H_2_ gas recovered the cells from oxidative stress. Various indicators including reactive oxygen species (ROS), nitric oxide (NO), antioxidant enzymes (catalase, glutathione peroxidase), intracellular calcium, and mitogen-activated protein kinase (MAPK) signaling pathway were examined to analyze the redox profile. The viability of A549 and NCI-H292 cells and the activity of antioxidant enzymes were reduced following induction by H_2_O_2_ and LPS but were later recovered using H_2_ gas. Additionally, the levels of oxidative stress markers, including ROS and NO, were elevated upon induction but were attenuated after treatment with H_2_ gas. Furthermore, H_2_ gas suppressed oxidative stress-induced MAPK activation and maintained calcium homeostasis. This study suggests that H_2_ gas can rescue airway epithelial cells from H_2_O_2_ and LPS-induced oxidative stress and may be a potential intervention for airway diseases.

## 1. Introduction

Oxidative stress indicates an imbalance of reactive oxygen species (ROS) and antioxidants, leading to disturbance of redox homeostasis and cellular damage. Oxidative stress triggers the pathological conditions in various disorders including pulmonary diseases [1,2]. Pulmonary epithelial cells are always exposed to varieties of stress inducers, and are the main target of ROS. In airway epithelial cells, ROS and their reactions participate in the pathophysiology of several pulmonary diseases, including asthma, pulmonary fibrosis, and respiratory distress, and stimulate impairment in pulmonary function, airway remodeling, and mucus secretion [3,4]. The pulmonary epithelial cells are protected in normal conditions by high intracellular and extracellular antioxidant levels [5].

Airway epithelial cells including A549 cells help to transport the substances (i.e., water and electrolytes) across alveoli. In addition, A549 cells reduce surface tension, which prevents alveolar collapse during respiration [6]. Similarly, NCI-H292 cells produce mucus and proteins in the bronchi, which forms a tight barrier for immune response [7]. Studies have shown that hydrogen peroxide (H_2_O_2_) and lipopolysaccharide (LPS) can be used as a strong oxidizing agent to induce oxidative stress and inflammation. With exposure to these compounds, alveolar and bronchial epithelial cell line such as A549 and NCI-H292 respond sensitively in lung injury due to oxidative stress [8,9,10]. One study showed that with the exposure of H_2_O_2_ in A549 cells, increased intracellular calcium (Ca^2+^) concentrations lead to mitotic arrest and cell apoptosis [8]. In addition, another study revealed that LPS activates endoplasmic reticulum stress and induces apoptosis in human airway NCI-H292 cells, resulting in acute lung injury [10].

In pathological and physiological conditions, mitochondria are mainly responsible for the production of ROS [11]. About 2% of ROS are emitted out of the entire oxygen utilized by mitochondria. Abnormality in production of ROS has a deleterious effect in proper functioning and regulation of cells [12]. Similarly, reactive nitrogen species (RNS), which includes nitric oxide (NO) and its derivative the peroxynitrite (ONOO-), acts as a powerful oxidant that can damage many biological molecules [13]. Moreover, cells bring defensive systems of antioxidants into active form, which is based mainly on enzymes such as glutathione peroxidase (GPx) and catalase (CAT), to defend from damage of cells caused by ROS [14]. Furthermore, mitogen activated protein kinase (MAPK) signaling is activated in response to various cellular stimuli. Three major subfamilies of MAPK are c-Jun N-terminal kinase (JNK), extracellular signal-regulated kinase (ERK), and p38. These can promote cell death by stimulating stress, including oxidative stress; thus, inhibiting these markers can increase survival [15].

With this, there is no doubt that oxidative stress plays a vital role in the progression of airway diseases. As a result of this, research interest to assess adjuvant effects of strong antioxidant agents has been increasing recently in order to alleviate these issues. Studies have shown that molecular hydrogen (H_2_) is a physiologically regulated gas molecule, which exerts antioxidant, anti-inflammatory, anti-apoptotic, and signal regulating properties [16,17]. Due to the fact that H_2_ can neutralize and convert highly active oxidants like the hydroxyl radical (^•^OH) and ONOO- into water, H_2_ is found to be beneficial for human health. H_2_ specifically quenches the ROS, while maintaining the metabolic oxidation-reduction reaction in the cell and can easily target organelles, including mitochondria and nuclei [16,18]. Several studies have reported that H_2_ suppresses oxidative stress-induced injury in various organs, such as brain, liver, and heart with minimal toxicity [19,20,21]. One of the previous studies reported that H_2_ protected cells against ROS and apoptotic damage induced by irradiation of cultured epithelial cells [22].

In the present study, we attempt to explore the antioxidative effect of H_2_ gas in alveolar and bronchial epithelial cells using A549 and NCI-H292, where oxidative stress is induced by H_2_O_2_ and LPS. Moreover, this study aims to determine the effect of H_2_ gas on oxidative stress via MAPK signaling pathway.

## 2. Materials and Methods

### 2.1. Cells and Chemicals

A549 and NCI-H292 cells utilized in this study were obtained from the Korean Cell Line Bank (Cancer Research Institute of Seoul National University, Seoul, Korea). The cells were cultured in cell culture flasks containing Roswell Park Memorial Institute (RPMI)-1640 medium, 10% fetal bovine serum (FBS) provided by Hyclone Laboratories (GE Healthcare Life Sciences, South Logan, UT, USA), and 1% antibiotic-antimycotic provided by Gibco (Life Technologies Corporation, NY, USA) at 37 °C with 5% CO_2_ and 100% humidity. H_2_O_2_ (Daejeong Reagent Manufacturer, Daejung, South Korea) and LPS from *Escherichia coli* O111:B4 (Sigma-Aldrich, Darmstadt, Germany) were used to induce oxidative stress.

### 2.2. Experimental Design

Oxidative stress was induced in A549 and NCI-H292 cells for 24 h using H_2_O_2_ and LPS. The cells were divided into four groups: normal control (NC; *n* = 3), induction with H_2_O_2_ or LPS only (Ind; *n* = 3), induction and treatment with H_2_ gas for 30 min (Ind + H_2_-30; *n* = 3), and induction and treatment with H_2_ gas for 60 min (Ind + H_2_-60; *n* = 3). In particular, A549 and NCI-H292 cells were cultured and sub-cultured in RPMI medium for a specific period of time. Both cells were isolated, counted, and seeded for 24 h, when the confluence of cells in the culture media reached approximately 80%. After the 24 h incubation period following the previous study by Begum et al. [23], the cells were washed, then the serum containing media was replaced with serum free media for cell synchronization and incubated with H_2_O_2_ (50 µM) or LPS (50 µg/mL), and then incubated for another 24 h at 37 °C with 5% CO_2_. After incubation, the cells were washed three times with 1× PBS and resuspended in RPMI medium. The relevant groups were treated with 2% H_2_ gas for 30 min or 60 min after which the cells were incubated for another 24 h. The 2% H_2_ gas (*v*/*v*%) was used in this study in reference to other several studies [24,25,26]. The cells were then lysed and the supernatant collected. The required part of the supernatant was used for enzymatic tests whereas the rest was stored at −80 °C for Western blotting assays of proteins in the MAPK pathway.

### 2.3. H_2_ Gas Treatment Procedure

The H_2_ gas treatment system was designed and provided by the company (GOOTZ Co., Ltd., Gyeonggi-do, Korea). Cell culture or multi-well plates were placed inside a panaqua cube (GOOTZ Co., Ltd., Suwon, Gyeonggi-do, Korea). A voltage stabilizer along with H_2_ timer (Dr’s Choice Co., Ltd., Tokyo, Japan) was used to fill the cube with H_2_ gas and measure the time. The cells were then exposed to H_2_ gas for either 30 min or 60 min. The H_2_ concentration in the panaqua cube was monitored using a cosmos XP-3140 (Combustible gas and vapor detector), as the H_2_ gas was being supplied to the cube (Figure 1). After treatment with 2% H_2_ gas for the specified time, the cell culture and multi-well plates were wrapped in aluminum foil. This foil aids in the long-term preservation of H_2_ gas and its activity in cultured cells [23,27]. The plates were then incubated for 24 h at 37 °C in the presence of 5% CO_2_ and then used for further experiments.

### 2.4. Cell Viability Assay

A Cell Counting kit-8 (CCK-8) reagent from Quanti-MaxTM (Seoul, Korea) was used to analyze cell viability and proliferation following the manufacturer’s instructions. In brief, A549 and NCI-H292 cells were seeded in a 96-well plate (2000 cells/well) with RPMI-1640 media and the plate was incubated at 37 °C with 5% CO_2_ for 24 h. After treatment with H_2_O_2_ and LPS, the media was discarded and the cells were washed and resuspended in RPMI media. Cells were treated with H_2_ gas (2%) for 30 min and 60 min and then incubated at 37 °C with 5% CO_2_ for 24 h. After 24 h of H_2_ treatment, 10 μL CCK-8 solutions were added to each well and the plate incubated at 37 °C for 2 h. A SpectraMax® ABS Plus (Molecular Devices, San Jose, CA, USA) was used to measure the optical density (OD) at 380 nm.

### 2.5. Intracellular Measurement of Total Reactive Oxygen Species (ROS)

The DCFDA Cellular ROS Detection Assay Kit (Abcam, Cambridge, MA, USA) was used to measure oxidative stress in A549 and NCI-H292 cells according to the manufacturer’s protocol. Briefly, 10 μL of the sample and 100 µL of 10 µM DCFH-DA were added into each well and the cells incubated for 30 min at 37 °C. Absorbance was measured at 488 nm excitation/525 nm emission using a DTX 880 multimode microplate reader (Beckman Counter Inc., Brea, CA, USA).

### 2.6. Quantification of Nitric Oxide (NO) Production

Griess reagent (Promega Corp., Madison, WI, USA) was used to quantify nitrite (NO_2_^−^) in A549 and NCI-H292 cells according to the manufacturer’s instructions. To evaluate the nitrate content, 50 μL of the sample was incubated at room temperature for 10 min with 50 μL of sulfanilamide solution. Next, 50 μL of NED solution was added and the mixture incubated for a further 10 min at room temperature. A SpectraMax® ABS Plus (Molecular Devices, San Jose, CA, USA) was used to measure the OD at 520 nm.

### 2.7. Assessment of Endogenous Antioxidant Enzyme Activity

Antioxidant enzyme activity was tested in A549 and NCI-H292 cell lysate following the manufacturer’s guidelines. Briefly, cells were plated in culture plates (1×10^6^ cells per well), and cell lysates were collected using assay buffer. The lysates were then centrifuged at 13,000× *g* rpm for 15 min at 4 °C. A SpectraMax® ABS Plus (Molecular Devices, San Jose, CA, USA) was used to quantify GPx activity (Biovision Inc., Milpitas, CA, USA) and CAT activity (Biomax Co., Ltd., Seoul, Korea) in the cell lysates. GPx absorbance was measured at 340 nm and CAT absorbance was measured at 560 nm.

### 2.8. Detection and Quantification of Intracellular Ca^2+^

Intracellular Ca^2+^ was measured in A549 and NCI-H292 cell lysates using a Ca^2+^ colorimetric assay kit (Biovision, Milpitas, CA, USA), according to the manufacturer’s instructions. In brief, cells were seeded in 6-well plates for 24 h. Following treatment, 50 μL of lysate was added to each well, along with 90 μL of chromogenic reagent and 60 μL of Ca^2+^ assay buffer. Absorbance was measured at 590 nm using a SpectraMax® ABS Plus (Molecular Devices, San Jose, CA, USA) after 10 min of incubation. The results are presented in mg/dL.

### 2.9. Quantification of Western Blot

Proteins for Western blot were extracted using RIPA buffer on ice. After protein estimation and normalization, cell lysates of cell were loaded in the gel and subjected to sodium dodecyl sulfate-polyacrylamide gel electrophoresis (SDS-PAGE). The gels were then transferred to the membranes. The membranes were blocked for 2 h at room temperature and incubated overnight with specific primary antibodies against p-p38, p-ERK, and p-JNK (dilution: 1:1000; Cell Signaling Technology, Massachusetts, Danvers, MA, USA) to observe the proliferation, stress response, inflammation, and cell apoptosis. The samples were then incubated with anti-rabbit secondary antibody (dilution: 1:5000; Cell Signaling Technology, Danvers, MA, USA) for 2 h at room temperature. ECL Western Blotting Substrate was used to detect the protein bands, which was then imaged using a UVP Biospectrum 600 Imaging System (UVP, LLC, Upland, CA, USA). The intensity of the protein bands was analyzed using Image J software.

### 2.10. Statistical Analysis

The mean ± standard error of the mean (SEM) were used to express the data. Analysis was conducted in GraphPad Prism 5.0 software (Graph-Pad, San Diego, CA, USA). Mean values of the groups were evaluated and assessed using one-way analysis of variance (ANOVA) followed by a multiple comparison test (Tukey post hoc test). Statistical significance was defined as a difference of *p* < 0.05.

## 3. Results

### 3.1. Augmentation of Cell Viability Post-Treatment of A549 and NCI-H292 Cells Using 2% H_2_ Gas

To evaluate the post-treatment effect of H_2_ gas on airway epithelial cells, A549 and NCI-H292 cells were treated with either 50 µM H_2_O_2_ or 50 µg/mL LPS for 24 h before treatment with 2% H_2_ gas (Appendix A). Our results showed that H_2_O_2_ induction reduced the cell viability in the A549 cell line compared with the control (*p* < 0.001), but this effect was significantly increased after treatment with H_2_ gas. LPS induction also decreased the cell viability (*p* < 0.001), but this was significantly improved, in a time-dependent manner, after treatment with H_2_ gas. Similar results were observed with the NCI-H292 cells. These results show that the reduced cell viability caused by H_2_O_2_ and LPS induction significantly increased, in a time dependent manner, upon treatment with H_2_ gas (Figure 2).

### 3.2. H_2_ Gas Attenuated Increased ROS and NO in A549 and NCI-H292 Cells

To demonstrate the antioxidant property of H_2_ gas, the levels of ROS and NO in both cell lines were determined following H_2_O_2_ and LPS induction. H_2_O_2_ and LPS induced ROS and NO production in both cell lines in comparison to untreated cells. In particular, H_2_O_2_ and LPS treatment significantly up regulated cellular ROS in the A549 cell line compared with the control (*p* < 0.001). The increase in ROS was reduced in a time-dependent manner after treatment with 2% H_2_ gas (Figure 3A). Similarly, when ROS production in NCI-H292 cells was significantly increased by induction with H_2_O_2_ and LPS (*p* < 0.001), H_2_ treatment significantly attenuated this increase in a time-dependent manner (Figure 3B). On the other hand, we observed a time-dependent reduction in NO levels in A549 cells following H_2_ gas treatment of H_2_O_2_ and LPS-induced cells (Figure 3C). Similarly, NO levels in NCI-H292 cells increased significantly following H_2_O_2_ and LPS induction, and this increase was attenuated in a time-dependent manner by treatment with H_2_ (Figure 3D). This finding indicates a time-dependent decrease in ROS and NO levels following treatment with H_2_ gas.

### 3.3. H_2_ Gas Has Beneficial Effect on Endogenous Antioxidant Enzyme Activities

Of the antioxidant machineries in the cell, we selected CAT and GPx to investigate the effect of H_2_ gas on the activities of these enzymes under oxidative stress. In A549 cells, inducing stress using H_2_O_2_ and LPS significantly increased CAT activity (*p* < 0.001). This increase in CAT activity was attenuated in a time-dependent manner by an exposure to 2% H_2_ gas (Figure 4A). Similarly, H_2_ gas treatment also significantly reduced CAT activity after H_2_O_2_ and LPS induction in NCI-H292 cells (Figure 4B). In contrast, the GPx assay showed a significant reduction in GPx activity after H_2_O_2_ and LPS induction in A549 cells (*p* < 0.001), and this was significantly rescued in a time-dependent manner after treatment with H_2_ gas (Figure 4C). Moreover, a statistically significant increase in GPx activity was observed in NCI-H292 cells after H_2_ gas treatment of H_2_O_2_ and LPS-induced cells (*p* < 0.001) (Figure 4D). These data indicate that treatment with H_2_ gas treatment mediates anti-oxidant enzyme activity under oxidative stress in a time-dependent manner.

### 3.4. Effect of H_2_ Gas on Quantities of Intracellular Ca^2+^ Concentration in A549 and NCI-H292 Cells

Oxidative stress alters Ca^2+^ homeostasis in the cells [28]. To determine the effect of H_2_ gas on intracellular Ca^2+^ concentration, A549 and NCI-H292 cells were exposed to similar oxidative stress-inducing agents. H_2_O_2_ stimulation increased intracellular Ca^2+^ concentrations in A549 cells compared with controls (*p* < 0.001), but this effect was significantly reduced in a time-dependent manner after treatment with H_2_ gas (Figure 5A). Intracellular Ca^2+^ levels in NCI-H292 cells considerably increased after exposure to H_2_O_2_ (*p* < 0.001) but reduced after treatment with H_2_ gas (*p* < 0.001). Furthermore, a time-dependent reduction in intracellular Ca^2+^ levels was observed (*p* < 0.001) when cells were stimulated with LPS and then treated with H_2_ gas (Figure 5B). These results demonstrate that intracellular Ca^2+^ was rescued in a time-dependent manner after treatment with H_2_ gas under oxidative stress.

### 3.5. Post-Treatment Effect of H_2_ Gas on the MAPK Pathway in A549 and NCI-H292 Cells

To investigate the mechanism of action of H_2_ gas in oxidative stress, p-JNK, p-ERK, and p-p38 proteins in the MAPK pathway were examined. In A549 cells, the results showed that upon H_2_O_2_ induction, levels of the MAPK pathway proteins increased significantly compared with controls (p-JNK (*p* < 0.001), p-ERK (*p* < 0.01), and p-p38 (*p* < 0.001)). Similarly, compared with controls, significant increase was observed in p-JNK (*p* < 0.01), p-ERK (*p* < 0.01), and p-p38 (*p* < 0.05) in A549 cells after LPS-induction. However, the increase in MAPK pathway proteins was attenuated following 30-min or 60-min exposures to H_2_ gas (Figure 6A).

In NCI-H292 cells, results revealed that after H_2_O_2_ induction, the MAPK pathway proteins increased significantly compared with the controls (p-JNK (*p* < 0.01), p-ERK (*p* < 0.01), and p-p38 (*p* < 0.001)). Similarly, compared with controls, a significant increase in p-JNK (*p* < 0.05), p-ERK (*p* < 0.01), and p-p38 (*p* < 0.01) was observed in NCI-H292 cells following induction with LPS. However, a decline in the expression of MAPK proteins in NCI-H292 cells was observed after treatment with H_2_ gas for specific time intervals (Figure 6B). These results demonstrate that H_2_ gas rescues the increase in MAPK proteins in a time-dependent manner.

## 4. Discussion

This study investigated the anti-oxidative effects of H_2_ gas on an airway epithelial cell lines. Several studies have shown that H_2_ gas has beneficial effects against oxidative stress through its enhancement of antioxidant levels in different cell lines [29,30,31,32,33]. In addition, oxidative stress plays a role in lung injury [34,35,36]. The safe and effective levels for H_2_ gas treatment are in the 1–4% range, whereas 4–75% represent explosive levels [32]. Yu et al. reported in 2017 that by using 2% H_2_, *v/v*% alleviates oxygen toxicity in PC12 cells by reducing hydroxyl radical levels. Similarly, another study conducted by Zhou et al. reported that exposing 2% H_2_ gas to AR42J cells significantly reduces the inflammatory cytokines level and oxidative damage. In addition to this, the safety of using 2% H_2_ has was also proved through the work of Ohta et al. [24,25,26]. Moreover, airway epithelial cells including A549 are alveolar cells, which prevent alveolar collapse by reducing surface tension, and NCI-H292 cells, which are located in the bronchi, act as a primary barrier for infection by secreting mucus and proteins [6,7]. Here, our study revealed the beneficial effect of 2% H_2_ gas on alveolar (A549) and bronchial (NCI-H292) epithelial cells via its inhibition of MAPK signaling pathway proteins. Specifically, our research focused on the anti-oxidative effects of 2% H_2_ gas on cell viability, ROS, NO, antioxidants, Ca^2+^, and MAPK signaling pathway proteins on airway epithelial cell lines at specific times and concentrations.

First, we investigated the effect of H_2_ gas treatment on cell viability after exposure to H_2_O_2_ and LPS. Several studies utilized H_2_O_2_ and LPS to simultaneously stimulate cellular oxidative stress and inflammation [37,38]. H_2_O_2_ produces O_2_^−^ by stimulating NADPH oxidase whereas LPS is detected by cells and activates NADPH oxidase and nuclear factor kappa B (NF-κB), both of which cause inflammation [38,39]. As expected, we observed that induction of A549 and NCI-H292 cells using H_2_O_2_ and LPS reduced cell viability. However, the administration of H_2_ gas resulted in a substantial time-dependent increase in the viability of both cell lines. These findings support our hypothesis that treatment with 2% H_2_ gas reduces H_2_O_2_ and LPS-induced cell death in bronchial and alveolar epithelial cells.

Similarly, ROS and RNS are essential regulatory mediators of the signaling process and are important for maintaining proper cellular activity at low concentrations [19]. Increased generation of ROS and NO can result in instability of the cellular redox state and irreversible and detrimental damage to the cell, resulting in cell apoptosis [40,41]. Thus, we assessed ROS and NO production in H_2_O_2_ and LPS-stimulated airway epithelial cells after H_2_ gas treatment. We showed that H_2_ gas treatment inhibited the increase in ROS and NO levels caused by H_2_O_2_ and LPS stimulation in airway epithelial cells. We also showed that a longer duration of H_2_ treatment was more effective in reestablishing ROS and NO balance. H_2_ gas diffuses quickly over the cell membrane and reaches the cell organelles, ultimately reducing the levels of free radicals [42]. These results are supported by the studies, which found that H_2_ gas could inhibit the detrimental effect of ROS and free radicals generated during lung injury [43,44]. These findings suggest that the harmful effects of excessive production of ROS and NO in alveolar and bronchial epithelial cells can be mitigated using H_2_ gas.

Moreover, antioxidant enzymes such as CAT and GPx, which selectively scavenge distinct types of ROS, help in ROS detoxification. H_2_O_2_ is converted to water and oxygen by CAT. GPx also aids in catalyzing H_2_O_2_ and organic hydroperoxides [45,46]. CAT and GPx enzymes help cells get rid of free radicals [47]. Consistent with the results of these studies, our findings revealed the effect of exposure to H_2_ at different time intervals on boosting the activity of antioxidant enzymes such as CAT and GPx after stress induction by H_2_O_2_ and LPS. Furthermore, cell injury results in the loss of Ca^2+^ and can be reversible or irreversible depending on the level of toxicity [48]. Thus, in this experiment, we used H_2_O_2_ and LPS to induce cell toxicity and observed an increase in Ca^2+^ concentrations. This increase in Ca^2+^ concentration was minimized when the cells were treated with H_2_ gas. This result is also supported by the studies, which found that H_2_O_2_ increases intracellular Ca^2+^ overload, and H_2_ might regulate the levels by modifying free radicals [49,50].

Finally, we focused on the MAPK signaling pathway to better understand the post-treatment mechanism of H_2_ gas-mediated oxidative stress. Several studies have demonstrated that MAPK proteins are essential for the conversion of extracellular signals into cell responses. The MAPK signaling pathway, which contains JNK, ERK, and p38 proteins, is one of the three primary pathways involved in cellular function. JNK and p38, commonly known as stress kinases, are activated simultaneously in response to a variety of cellular and environmental stresses, whereas ERK 1/2 is activated in response to growth stimuli [40,51,52,53]. These three pathways were investigated with reference to H_2_O_2_ and LPS stress induction in airway epithelial cells. We observed that H_2_ gas reduced H_2_O_2_ and LPS-induced MAPK signaling pathway proteins, JNK, ERK, and p38, in a time dependent manner. These results provide evidence of the beneficial role of H_2_ gas in H_2_O_2_ and LPS-induced oxidative stress via the MAPK signaling pathway. Taken together, treatment with H_2_ gas rescued or restored the oxidative stress to pre-induction levels, which varied depending on the cell lines, induction method, and time for H_2_ gas treatment. Moreover, a 60-min H_2_ gas treatment was found to be more effective than a 30-min treatment in this study.

## 5. Conclusions

In conclusion, H_2_ gas protected bronchial and alveolar epithelial cells against oxidative stress. Moreover, in both cell lines, exposure to H_2_ gas for 60 min was more effective than exposure for 30 min. Oxidative stress markers and proteins in the MAPK pathway were altered after induction with H_2_O_2_ and LPS. These changes were alleviated by the administration of H_2_ gas. The findings provide an effective way for reducing oxidative stress-induced injury in bronchial and alveolar epithelial cells. However, further studies are needed to fully elucidate the molecular mechanisms underlying H_2_ function in vivo and clinical studies. However, to our knowledge, this is the first report that shows the antioxidant effect of H_2_ in both bronchial and alveolar epithelial cells in H_2_O_2_ and LPS-induced oxidative stress.

## Figures and Tables

**Figure 1 molecules-26-06375-f001:**
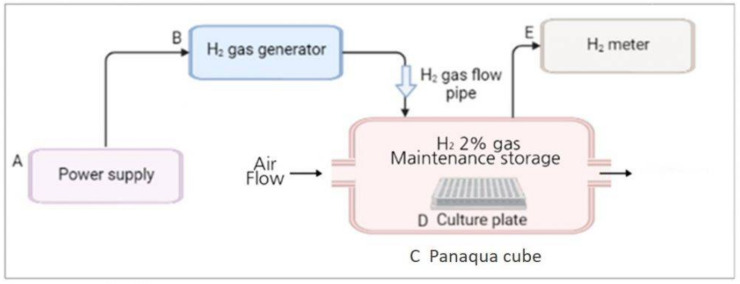
Overall schematic representation of the H_2_ gas treatment process. (**A**) Power supply was set at around 9 volt; (**B**) H_2_ gas generator (produces 2% H_2_ gas); (**C**) Panaqua cube (contains culture plate for supplying H_2_ gas); (**D**) Cell culture plate; (**E**) H_2_ gas meter (measures the concentration of H_2_ gas in the cube).

**Figure 2 molecules-26-06375-f002:**
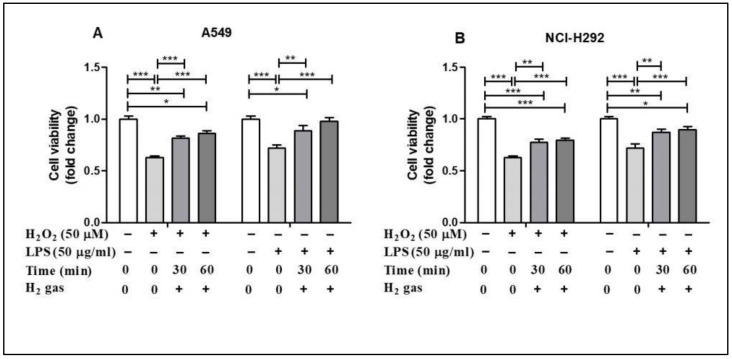
Effect of 2% H_2_ gas on cell viability in A549 and NCI-H292 cells induced with H_2_O_2_ and LPS. Effect of H_2_ gas (30 min and 60 min post-treatment) on the viability of (**A**) A549 cells induced with H_2_O_2_ (50 µM) and LPS (50 µg/mL) and (**B**) NCI-H292 cells induced with H_2_O_2_ (50 µM) and LPS (50 µg/mL). Data are presented as mean ± SEM. * *p* < 0.05, ** *p* < 0.01 and *** *p* < 0.001 represent significant differences based on ANOVA.

**Figure 3 molecules-26-06375-f003:**
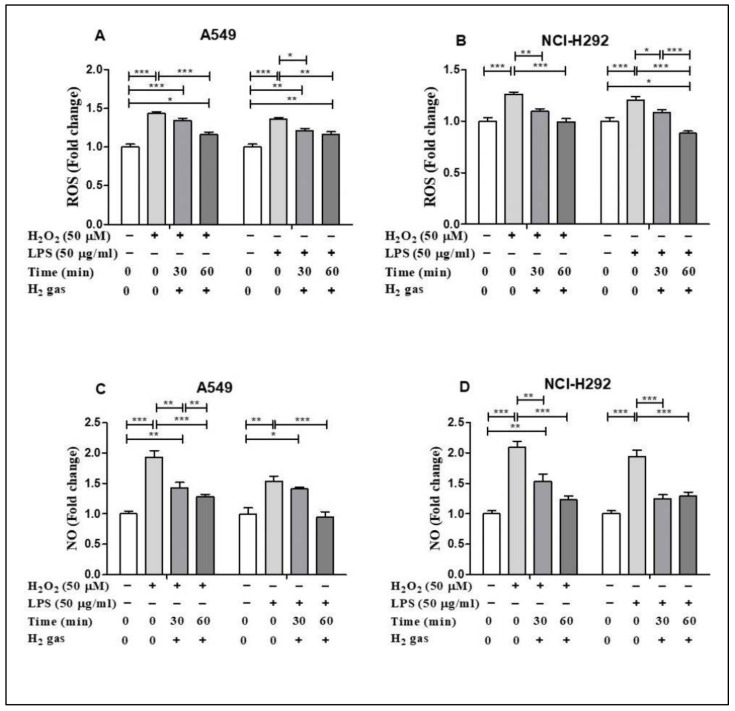
Effect of 2% H_2_ gas on ROS and NO production in H_2_O_2_ and LPS-induced A549 and NCI-H292 cells. Cells were induced with H_2_O_2_ (50 µM) or LPS (50 µg/mL) for 24 h, treated with H_2_ gas for 30 min or 60 min, and then incubated for another 24 h. The supernatant was collected and used to perform ROS and NO assays. (**A**) ROS levels in A549 cells. (**B**) ROS levels in NCI-H292 cells. (**C**) NO levels in A549 cells. (**D**) NO levels in NCI-H292 cells. Data are presented as mean ± SEM. * *p* < 0.05, ** *p* < 0.01 and *** *p* < 0.001 represent the level of significance based on ANOVA.

**Figure 4 molecules-26-06375-f004:**
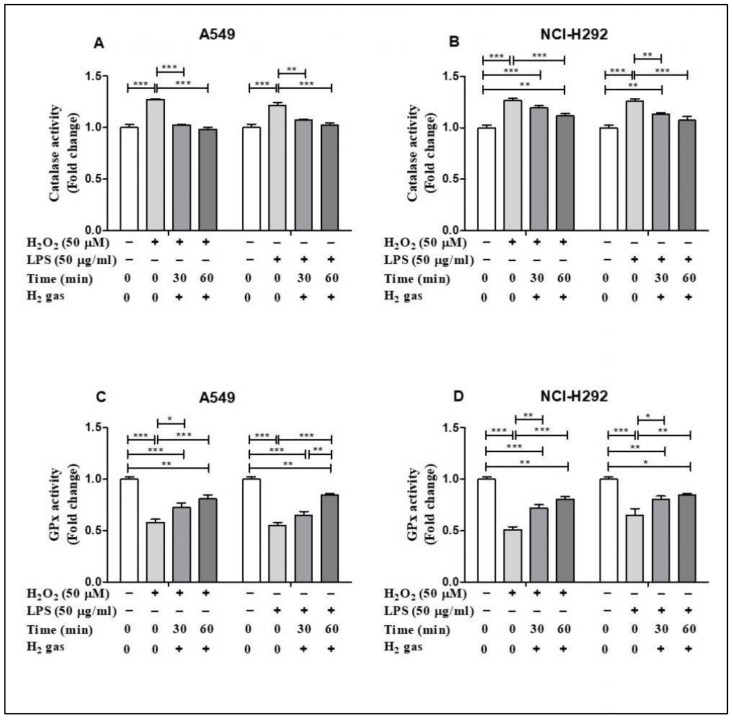
Influence of 2% H_2_ gas on CAT and GPx activity in H_2_O_2_ and LPS-induced A549 and NCI-H292 cells. H_2_O_2_ and LPS were administered to cells at concentrations of 50 µM and 50 µg/mL, respectively. RPMI media was added and the cells incubated for 24 h after which they were washed and rinsed with 1x PBS. After 24 h of H_2_ treatment, the supernatant was isolated and CAT and GPx assay were performed. (**A**) CAT activity in A549 cells. (**B**) CAT activity in NCI-H292 cells. (**C**) GPx activity in A549 cells. (**D**) GPx activity in NCI-H292 cells. Data are presented as mean ± SEM. Statistical significance was computed using ANOVA. * *p* < 0.05, ** *p* < 0.01 and *** *p* < 0.001.

**Figure 5 molecules-26-06375-f005:**
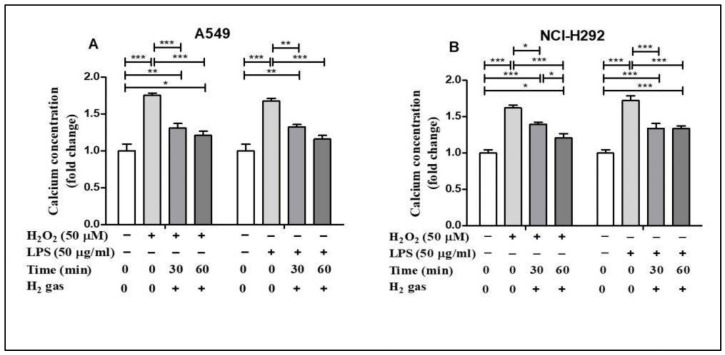
Effect of 2% H_2_ gas on intracellular Ca^2+^ levels in H_2_O_2_ and LPS-induced A549 and NCI-H292 cells. Cells were treated with 50 µM of H_2_O_2_ or 50 µg/mL of LPS and incubated for 24 h. The cells were then washed and the media replaced with normal RPMI. The supernatant was collected after 24 h of H_2_ treatment and a Ca^2+^ assay was performed. Ca^2+^ concentrations were measured in (**A**) A549 cells and (**B**) NCI-H292 cells using one way ANOVA. * *p* < 0.05, ** *p* < 0.01, and *** *p* < 0.001 represent the significance levels of the results.

**Figure 6 molecules-26-06375-f006:**
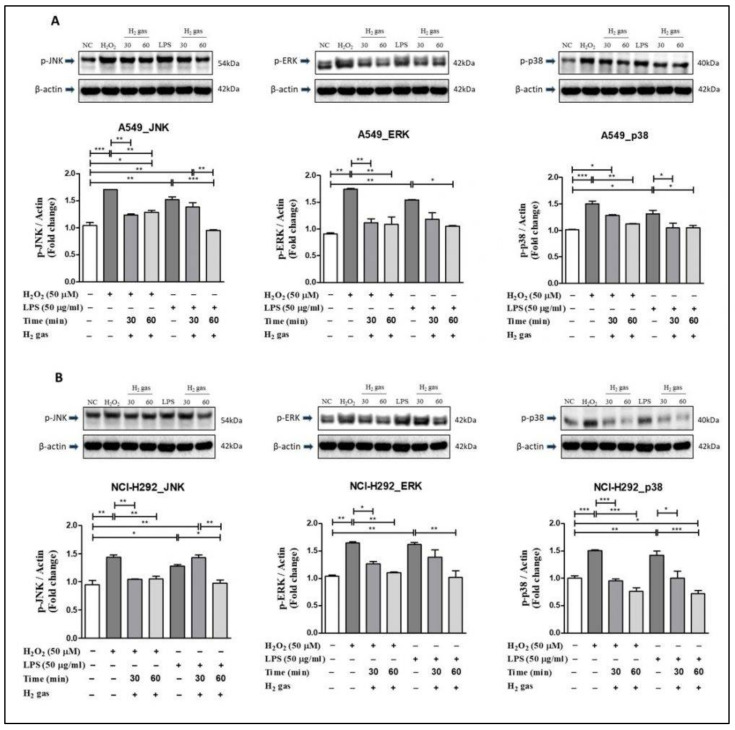
Effect of 2% H_2_ gas on MAPK pathway proteins in cells treated with H_2_O_2_ and LPS. MAPK proteins p-JNK, p-ERK, and p-p38 were analyzed following H_2_ treatment of (**A**) A549 cells; and (**B**) NCI-H292 cells and compared with untreated controls. Image J was used to analyze the data. Data are presented as mean ± SEM. * *p* < 0.05, ** *p* < 0.01 and *** *p* < 0.001 represent the level of significance based on ANOVA.

## Data Availability

All the data are contained within the article.

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
