# Peer review of "Antioxidant Properties of Hydrogen Gas Attenuates Oxidative Stress in Airway Epithelial Cells"

_molecules, 2021, doi:10.3390/molecules26216375_

Round 1

Reviewer 1 Report

Reviewer Comment

The authors investigated the antioxidant activity and mechanism of H2 gas on two types of cultured lung cancer cells, A549 and NCI-H292. The results showed that H2 gas improved the survival ratio, suppressed the productions of ROS and NO, and inhibited the decrease in antioxidant enzymes, increase in intracellular calcium concentration and the activation of MAPK. Since these findings suggest the efficacy of H2 gas against airway diseases and its possible mechanisms, I think the paper will be informative for the readers of Molecules. However, I think the following revisions are necessary.

1. Since 100% H2 gas dissolves 1.6 ppm (mg/L) in the culture medium, the concentration of H2 in the culture medium using 2% H2 gas in this experiment is 0.032 ppm. Since Ohsawa et al. treated cultured cells with 75% H2 gas in their paper Nature Medicine, 13: 688-694, 2007, it means that the experimental conditions used in this study were very low concentration of H2 gas. It would help the readers of this paper to understand if the authors cite other references that used 2% H2 gas in cell culture experiments and comment that the antioxidant effect of H2 gas can be studied even under the present experimental conditions.

2. Please show the reader in an easy-to-understand manner with a figure of the mechanism regarding the antioxidant activity of H2 gas assumed from the results of this experiment.

3. Please correct Figure 3B: H292 to NCI-H292.

Author Response

The authors really appreciate the reviewer for his careful review and feedback. We have attached the file which addressed your comments and point to point response for your comments. Thankyou so much for your time.

Reviewer 2 Report

In their article, the authors investigate the role of H2 gas on the oxidative profile of two airway cell lines (alveolar A549 and bronchial NCI-H292). The results indicate that H2 gas treatment dose-dependently improves the REDOX profile in both cell types and opens up a rationale for using H2 gas in treating respiratory diseases.

The article is well written; both the experimental design and the results, discussion and conclusions are correct and interesting from a clinical point of view. In my opinion, the article is suitable for publication, provided that the authors answer the following questions:

1.- In my opinion, the chapter on materials and methods should appear after the introduction and before the results. The methods are as important as or more important than the results for a good evaluation of research work. I believe that readers will appreciate this change.

2.- Why are two cell lines used? Is there any reason beyond their anatomical origin for using two cell lines? Please explain.

3.- Why was a 24 hr incubation time used? Were time-course experiments performed? In that case, please, provide evidence in supplementary materials.

4.- In the experimental design section, line 263, change H2O2/LPS for H2O2 or LPS because it leads to thinking that experiments with both agents have been performed simultaneously, which is indeed clarified further down in the text (line 268). Still, it confuses the first time it is read.

5.- In line 268, the authors indicate that they synchronize the cells. How and why do they synchronize them? Please explain.

6.- In line 284, the authors indicate that "... multi-well plates were wrapped in aluminium foil". Please explain why?

7.- In the results chapter, the authors do not explain the differences between the control and induced + treated groups. The question is, does the treatment reduce oxidative stress to pre-induction levels, i.e. to normal levels?

Author Response

The authors really appreciate the reviewer for his careful review and feedback. We have attached file below which addressed your comments and point to point response towards those comments. All the authors of this manuscript thank you for your valuable feedback and time.

Round 2

Reviewer 1 Report

Reviewer's comments

I confirm that accurate corrections have been made to my comments. I recommend that this paper be accepted.

Reviewer 2 Report

Dear authors:

Many thanks for the thorough review and for answering my comments. I think that the article is ready to be published in its current form.